# Characterization of Pharmaceutical Tablets by X-ray Tomography

**DOI:** 10.3390/ph16050733

**Published:** 2023-05-11

**Authors:** Jaianth Vijayakumar, Niloofar Moazami Goudarzi, Guy Eeckhaut, Koen Schrijnemakers, Veerle Cnudde, Matthieu N. Boone

**Affiliations:** 1Centre for X-ray Tomography (UGCT), Ghent University, Proeftuinstraat 86/N3, 9000 Gent, Belgium; 2Department of Physics and Astronomy, Radiation Physics, Ghent University, Proeftuinstraat 86/N12, 9000 Gent, Belgium; 3Janssen Pharmaceutica, Turnhoutseweg 30, 2340 Beerse, Belgium; 4Pore-Scale Processes in Geomaterials Research (PProGRess), Department of Geology, Ghent University, Krijgslaan 281/S8, 9000 Gent, Belgium; 5Environmental Hydrogeology, Department of Earth Sciences, Faculty of Geosciences, Utrecht University, Princetonlaan 8A, 3584 CD Utrecht, The Netherlands

**Keywords:** X-ray tomography, pharmaceutical tablets, image processing, X-ray tomography modalities

## Abstract

Solid dosage forms such as tablets are extensively used in drug administration for their simplicity and large-scale manufacturing capabilities. High-resolution X-ray tomography is one of the most valuable non-destructive techniques to investigate the internal structure of the tablets for drug product development as well as for a cost effective production process. In this work, we review the recent developments in high-resolution X-ray microtomography and its application towards different tablet characterizations. The increased availability of powerful laboratory instrumentation, as well as the advent of high brilliance and coherent 3rd generation synchrotron light sources, combined with advanced data processing techniques, are driving the application of X-ray microtomography forward as an indispensable tool in the pharmaceutical industry.

## 1. Introduction

Pharmaceutical drugs can be administered using a wide variety of dosage forms, ranging from solid (tablets, capsules, powders), liquid (injections, solutions, emulsions, lotion, suspensions, creams, ointments) and gas (sprays, vaporizers, aerosols, nebulizers, atomizers or inhalers). The solid dosage forms, especially tablets and capsules are widely used as they are simple to administer, have higher stability (less dependency on temperature/pressure), are easy to handle for shipping and logistics, can be easily designed/optimized for well controlled drug release and can be consistently produced in large volumes at relatively low costs with minimal waste [1,2,3,4]. A solid dosage form of a drug consists of two parts: the active pharmaceutical ingredients (API) and inactive ingredients/excipients. The APIs consist of the medicine(s) used for the treatment while the inactive ingredients may consist of one or more chemical compounds such as binders, filler or coloring agents to name a few. The development of a solid dosage form starts with the step known as “preformulation testing” [5,6]. Based on the preformulation testing, in addition to the API, excipients are selected depending on the type of formulation. The API grains are typically converted into a suitable size as per the dosage requirement. Then, all the ingredients are either blend together to obtain a uniform mixture, or granulated to facilitate a good mixing. A tablet can be made by a multitude of techniques, such as compression, 3D printing or freeze drying to name a few. In some cases the powder form can also be introduced into a capsule. The formulation and manufacturing process of the tablet are designed to optimize the systematic release of the API in order to achieve the desired therapeutic effect.

Drug absorption from a tablet after administration depends on the release of the APIs from the drug product, such as the dissolution or solubilization of the APIs under physiological conditions, and the permeation across various entry points such as the gastrointestinal membrane [7,8,9]. The macroscopic and microscopic morphology of different ingredients and their distribution are key in the functioning of the dosage form. Despite carrying out the preformulation tests, further analysis is needed to ensure the characteristics of the final tablet after the production, and to verify if the necessary functionality can be achieved [1,4,10,11,12]. The functionality of the tablet depends on (a) intrinsic characteristics such as the content distribution (APIs and excipients), concentration of pores, morphology of the granules, presence of impurities and solvent; (b) extrinsic characteristics related to mechanical strength and stability, and (c) the performance when ingested such as stability of the coating layer, the mechanism of drug release, and the dissolution process. Characterizing the intrinsic and extrinsic properties is needed to check the drug product quality and dissolution tests are needed to check the performance. Analyzing the intrinsic and extrinsic characteristics and their uniformity within a production series are important for the drug development, quality control to optimize the manufacturing process, and approval from drug regulatory agencies such as the Food and Drug Administration agency (FDA) in the USA or the European Medicines Agency (EMA) in Europe [13,14,15,16,17,18]. The acceptance criteria for a drug product are set to guarantee the quality and performance of the product over its lifetime. Specific guidelines are set for drug release and dissolution assessment of the final tablet and should be consistent with different batches of tablets when produced on a large scale [14,18,19,20]. Poor characterization may result in poor functionality of the tablet which can result in failing the drug approval/assessment tests and possible recalling of the tablets from the market if previously approved. In particular, issues with poor disintegration and dissolution have been a major factor in recalling the tablets which were already in circulation [20].

The final tablets are routinely checked during the production. For example, the identification and assay of the API, or residual solvent content is checked during the production, and the associated tests need to be carried out on a fast time window. A standardized characterization procedure to test the quality and performance is typically developed, and carried out in detail during the tablet development process, or in some cases on different batches during the production. Standard characterization tools in pharmaceutical industry typically include chemical analysis such as titration or chromatography methods. However, during the developmental stages, one can obtain the combined physical and chemical properties of the tablet through spectro-microscopy methods, which can be a valuable supplementary characterization for the development of the standardized and validated analysis package. Table 1 summarizes several commonly used evaluation methods carried out on the manufactured tablets during the development stage. In addition to the chemical analysis, standard analytical tools include UV spectroscopy/microscopy [21,22,23], mid/near infrared spectroscopy [24,25,26,27,28,29,30], Raman (microscopy and spectroscopy) [31,32,33,34,35], scanning electron microscopy + electron dispersive spectroscopy (SEM + EDS) [36,37,38,39,40]. Other techniques which are less used or being explored include transmission electron microscopy (TEM) [39,41], X-ray microtomography [20], magnetic resonance spectroscopy [42,43]/imaging [44,45] optical coherence tomography [20], THz microscopy/tomography [20,46], and synchrotron techniques such as small angle X-ray scattering (SAXS) [47], and X-ray diffraction [48,49,50]. Spectro-microscopy methods such as SEM or Raman microscopy can offer combined spatial and chemical resolution, however, the characterizations are often destructive in nature. For example, to characterize the homogeneity of the tablet contents, the tablet is cut before being imaged by Raman microscopy which is limited to only one surface.

Nevertheless, during the tablet formulation and development process, it is very valuable that the internal tablet characteristics are assessed without damaging the tablet for a direct comparison with standard techniques for better accuracy. Hence, non-destructive microscopy characterization of the final manufactured tablet became a necessity, and 3D imaging or computed tomography techniques could potentially be used. Possible non-destructive 3D imaging and tomography methods which can be used to characterize solid dosage forms include X-ray microtomography [20], magnetic resonance imaging/nuclear magnetic resonance (MRI/NMR) [44,45], transmission terahertz imaging/tomography [20,46], and optical coherence tomography [20]. Other possible approaches include confocal Raman microscopy [59,60], positron emission tomography [61] and neutron tomography [62]. For simplicity, we draw here a distinction between tomography imaging and 3D imaging. We consider tomography (or computed tomography) imaging as a computational imaging technique where a set of transmission images taken at different rotating angles are computationally reconstructed to obtain a 3D image. Common examples of this technique are X-ray microtomography or neutron tomography. However, for a 3D imaging the interaction between the probe and the material is mapped across the volume of the sample without employing a reconstruction procedure. Confocal microscopy or reflection mode THz pulse imaging can be classified under this category. A brief description of different 3D non-destructive imaging and tomography approaches suitable to characterize pharmaceutical tablets are listed as follows:X-ray microtomography. This increasingly popular technique can be carried out in lab-based as well as at synchrotron-based X-ray sources [63,64,65,66,67,68,69,70], and sub μm resolution can be achieved in both instruments. X-ray microtomography requires minimal to no additional sample preparation making it a simple and convenient technique for tablet characterization as well as for the in situ imaging. Standard modalities include the measurement of the absorption or phase contrast of the sample. However, local diffraction contrast or refractive indices can also be characterized by diffraction contrast tomography and holotomography, respectively. Chemical resolution is minimal, however can vary with the elemental contents of the pharmaceutical ingredients. Sub-10 nm resolution can be achieved by ptycho-tomography on smaller samples. X-ray microtomography, being the main topic of this manuscript, will be elaborated in the later sections.Magnetic resonance imaging/Nuclear magnetic resonance. The origin of magnetism in the atomic nucleus is the uneven number of protons and neutrons resulting in a net magnetic moment in the nucleus. NMR measures the interaction between the oscillating external magnetic field and the Larmor frequency of the nuclei (or atom in case of a magnetic material such as Fe, Ni or Co). Therefore, not all the elements can be detected using magnetic resonance imaging and may require contrast agents [45]. In the MRI setup the magnetic field is applied in different directions in a particular slice and the response is measured by the receiving coils. The responses from different orientations are combined to produce a 3D image, and a spatial resolution of 20–1000 μm can be achieved [20]. Spatially resolved NMR is usually termed as MRI. Porosity [71], density distribution [20,45,72], and dissolution of the tablet [44,73] can be characterized by MRI imaging. Furthermore, in situ studies on the drug release and absorption in human or animal subjects can also be carried out by MRI [45].Neutron tomography. While X-ray photons interact with electrons, neutrons interact with the atomic nucleus. Neutron tomography can be carried out by characterizing the neutron absorption [74], scattering cross-section [75,76], local diffraction [77] as well as by the phase change [78]. Neutrons are highly sensitive to light elements such as H, N, C, O and are attenuated very efficiently by proton-rich compounds such as water [79], and can be suitable to characterize in situ dissolution/disintegration of tablets. Spatial resolution below 5 μm can be achieved with chemical resolution [80,81]. Until now, neutron tomography has not been used in characterizing pharmaceutical dosage forms, as they become radioactive after the characterization [62].THz tomography/imaging. THz imaging or THz pulse imaging can be carried out in transmission or reflection mode depending on the type of material. The transmission is characterized by the frequency dependent absorption or phase shifts associated with the transmitted wave, and can be used as a tomography technique to produce a 3D image. The reflection mode is typically used as a 3D imaging tool, where the change in the THz pulse after reflection is spatially resolved across the reflecting surface. The interaction of a THz pulse with the material can be used to determine the local refractive index and interamolecular vibration modes/lattice dynamics. Hence, THz pulse imaging/spectroscopy can provide both physical and chemical information from the sample through spectroscopy measurements. The possibility to obtain both spatial and chemical information makes THz pulse imaging a suitable tool for non-destructive characterization of pharmaceutical tablets [20]. THz radiation have wavelength from 1 mm to 10 μm whose transmission and reflection depends on the thickness and dielectric constant of the material. A THz pulse can have a typical power in the order of a few μW and can probe a depth of 2 mm depending on the material composition. The technique can have a spatial resolution between 50–100 μm. Therefore, a non-destructive analysis similar to X-ray tomography is not yet possible [82]. Nevertheless, it has been demonstrated that on selected tablets, the porosity [83,84] and coating layer thickness can be effectively characterized by this approach [85].Optical coherence tomography. This is an interferometry imaging technique where the interference between the output laser beam from the source and the interacted laser beam from the sample is measured. The laser interaction depends on the refractive index of the material and the penetration depth, which alters the interference pattern due to the change in the coherence length. The changes measured from the interference pattern is mapped across the sample. The interacted light beam is typically acquired by measuring the light reflection. Typically, near infrared wavelength is used which can provide a spatial resolution in μm range and imaging depth in mm range is possible. Due to the low penetration depth of the infrared light, its application in pharmaceutical tablets is typically limited to the coating layer [86,87]. The suitability for other applications in pharmaceutical science is yet to be explored, however, with the existing tools the applicability is similar to that of THz imaging [20].Confocal Raman microscopy. In general, confocal microscopy measures the fluorescence or the reflectance from the sample. For better chemical resolution it can be combined with Raman spectrometry, where the phonon vibration mode of the molecules is measured. A lateral resolution of 100–200 nm and an axial resolution of 500 nm can be obtained [88]. Probing depth of around 100 μm can be reached using visible light however higher probing depth in the order of mm can be achieved by using an infrared wavelength and can vary with the optical properties of the sample. 2D Raman microscopy combined with microtome cutting capability can be used to obtain a chemically resolved 3D image, however, this process is destructive. While Raman spectroscopy or microscopy is largely used in the pharmaceutical industry, non-destructive 3D Raman microscopy to measure the entire tablet is not yet possible.

The choice of which non-destructive method to use depends on the type of characterization required. X-ray microtomography is one of the most versatile techniques, offering the best trade-off between sample preparation, sample size, resolution, and measurement time in both lab-based and synchrotron X-ray sources. However, due to the lack of direct chemical resolution, it is often difficult to classify the individual chemical components of the pharmaceutical ingredients on microtomography images. Therefore substitute chemical characterization such as SEM-EDX and/or Raman microscopy is still needed to support the X-ray microtomography data. Nevertheless, information on the particle distribution, thickness, size and shape of the segmented region can be accurately measured in 3D X-ray microtomography. Furthermore, with the availability of 3rd generation synchrotron sources, the development of detectors, and advanced computational algorithms, X-ray microtomography can offer unprecedented information on the internal structure of the solid dosage forms, making it a valuable tool for the pharmaceutical industry and research. In this article, we review in detail the applicability of X-ray microtomography to characterize selected intrinsic and extrinsic tablet attributes which are closely related to the quality and performance of the tablets. We discuss in detail the different modalities of X-ray tomography suitable for pharmaceutical industry, identify various software for image reconstruction, data analysis and provide an overview of the image analysis workflow. Thereupon, we review separately the different tablet characterization techniques to analyze the quality and performance, as well as describe the existing standardized methods, the corresponding quantifiers (such as porosity, mixing indices) and the applicability of X-ray microtomography along with the respective data analysis towards such characterization.

## 2. X-ray Microtomography

The X-ray energy range is typically considered between 100 eV and 200 keV, corresponding to wavelengths from nm to pm range. X-rays have played an important part in the characterization of materials for their ability to determine the physical, chemical, electronic and structural properties [89]. The interaction of X-rays with the material can result in attenuation or scattering of the X-ray beam. The extent of interaction is determined by the sample thickness, X-ray energy, incidence angle and refractive index of the material. The refractive index is dependent on the atomic scattering factor/form factor, and the former can be written as a complex term as n=1−δ+iβ. The magnitude of the imaginary part is called the absorption index and is proportional to the absorption coefficient, while the real part is called refractive index decrement which determines the refractive index of the material. A photo-absorption process results in the excitation of electrons from lower energy to higher energy levels [90]. The highest transition of electron excitation occurs when the X-ray energy matches with the binding energy of the electron (also known as the absorption edge), making it possible to identify elements, in its pure form as well as in its compound form [91,92]. The absorption of a material is determined by the mass absorption coefficient μ. It is proportional to the absorption cross section and inversely to the atomic weight ($\mu =\sigma_{A}N_{A}/A$, $N_{A}$ is the Avogadro’s number and A is the atomic weight). The absorption cross section ($\sigma_{A}$ [m^2^]) varies with X-ray energy and the type of scattering during the X-ray interaction with matter. The different types of scattering include elastic (Rayleigh), inelastic (Compton) scattering, ejection of electrons due to photoelectric effect and pair production such as electron and positron, or nuclear absorption. The former three effects occur in soft and hard X-ray range (100 eV–200 keV), while the latter occurs in higher energy range in the order of MeV depending on the material. The absorption coefficient can be experimentally determined from the intensity of the transmitted monochromatic X-ray, given by Beer-Lambert’s law,
(1)$$I=I_{0}e^{-\mu_{l}t}$$
where *I* is the transmitted intensity, $I_{0}$ is the X-ray intensity before transmission, $\mu_{l}$ is the linear attenuation coefficient, and *t* is the thickness. The linear attenuation coefficient is related to the mass attenuation coefficient μ as $\mu_{l}=\mu \cdot \rho$. This material-specific parameter depends strongly on the incident X-ray energy, and is tabulated [90]. The transmitted intensity typically reduces with the sample thickness, atomic number, and absorption cross-section of the sample. The X-ray transmission through the sample is typically measured by a 2D detector, where the resultant data would be an absorption image or a radiograph [an example is shown in Figure 1a]. The real term in the refractive index determines the wave propagation within the material, due to which the transmitted wave intensity is not only reduced due to the absorption but can also be out of phase with the incident wave. By resolving the phase difference across the sample, one can produce phase contrast images/radiographs, similar to absorption images and are particularly suitable for low-Z materials. Tomography is one of the imaging techniques where transmitted absorption images (i.e., radiography images) are acquired at different rotating angles, with the rotating axis perpendicular to the light propagating vector. From the radiography images, a 3D image with voxels (3D unit pixel) representing the local linear attenuation coefficient is produced by using a reconstruction algorithm [63,64,66].

X-ray microtomography is a versatile imaging technique and can be carried out using polychromatic lab-based as well as monochromatic synchrotron-based X-ray sources. A tomography setup typically consists of an X-ray source, a rotating stage (along with a linear translation stage) and a detector. The important differences between lab-based and synchrotron tomography setups are listed as follows;

Lab-based X-ray microtomography setup. The X-ray beam is generated by focusing a narrow electron beam on an anode material (such as W or Mo) with a high acceleration voltage. The resultant X-ray beam has a white spectrum, covering the full energy range up to the energy of the electrons, which is in the range of 30 to 200 keV. The X-ray beam produced in lab-based sources is typically isotropic, due to which the imaging geometry can be considered conical, allowing for the possibility of geometrical magnification of the sample by adjusting the source-sample and sample-detector distances. The resolution of the image is determined by the size of the X-ray beam spot probing the sample and the magnification [93,94]. Figure 1a shows the lab-based set up with cone beam geometry. A 2D array pixel detector, either based on indirect detection (i.e., using a scintillator screen to convert X-rays to visible light) or direct detection (i.e., directly converting the X-rays to electron-hole pairs in the semiconductor sensor), is used to acquire the radiography images.Synchrotron-based X-ray microtomography setup. The X-ray beam is generated by accelerating electrons close to the speed of light. The accelerated electrons emit collimated and polychromatic radiation in the forward direction of the electron motion. The photons produced as a result of acceleration of electrons are separated from the electrons by insertion devices such as a bending magnet which deflect the electrons from their path. The separated photons are then directed towards the sample for characterization. The X-ray beam in a synchrotron source is generated as a parallel beam with low divergence and high flux (hence high brilliance) [as shown in Figure 1b] and spatial coherence [89]. The X-ray beam is commonly monochromatized by a monochromator and is further aligned by using X-ray optics. The low divergence enables long propagation distances with both high spatial and high longitudinal (or temporal) coherence from the insertion device to the sample. Such high degree of coherence made other modalities of X-ray imaging such as phase contrast imaging possible. The X-ray beam size typically ranges between sub 100 μm to cm in diameter, and at X-ray microtomography beamlines, the diameter of the X-ray beam can range from a few mm to cm. With a parallel beam geometry, the resolution is determined by the detector resolution, i.e., the number of pixels and its area. A thin scintillator screen combined with optical magnification is typically used in synchrotron X-ray microtomography set-ups and the acquisition time is faster than lab-based set-ups with a better signal to noise ratio due to the high flux. The detection efficiency of a scintillating detector is typically low, however, due to the high flux of the synchrotron X-rays, a higher overall efficiency is achieved [95,96,97]. The combination of scintillator and an optical microscope is not only an effective way to magnify, but also overcomes the need to have a detector with very small pixel resolution. If the size of the object is larger than the X-ray spot, smaller radiography images are acquired and are later stitched together to form a full radiography image of the sample.

For standard tomography imaging, once the radiography images are acquired, the images are processed to subtract the background signal, normalize on the incident flux and possibly reduce noise and artefacts by filtering. For computational reasons, the images are typically converted to sinograms, which are the intensity profiles of each detector row at all the measured angles of rotation. The sinograms are then used to reconstruct the internal structure of the object or 3D absorption map, with each voxel representing the local attenuation coefficient [98]. The sinogram from each detector row is used to reconstruct a slice of the 3D object. The reconstruction is carried out as a multidimensional inverse problem, and can be solved by different reconstruction algorithms available such as back projection algorithm or iterative reconstruction algorithm [20,70,98,99,100]. The resolution of the reconstructed tomography image/slice is dependent on the resolution of the raw projection data, angular sampling (i.e., the number of projections per rotation), and instrument conditions such as object stability, geometrical misalignment, etc.

Throughout the years, different variations of X-ray microtomography techniques have been developed, aiming at extending the capabilities of conventional X-ray tomography, where local attenuation is reconstructed in a 3D space. The newer variations aim at increasing the spatial and/or chemical resolution, or providing additional information such as local refractive index or crystallinity. The different modalities of X-ray tomography are listed as follows and a short summary is given in Table 2;

Nanotomography—It is an extension of microtomography achieved through technical enhancements in the detector and X-ray sources. High resolution tomography is needed to characterize smaller features such as pores or small pharmaceutical particles present in the solid dosage forms. X-ray tomography with nm-scale resolution is classified as nanotomography.(a)Lab-based nanotomography. In lab-based set-ups, making the focal spot smaller enables to increase the geometrical magnification while keeping image sharpness, hence increasing the resolution. However, smaller focus spot result in lower flux making it difficult to measure thick/low dense samples. Sub-micron resolution (300 nm–1 μm) can still be achieved with such configuration (without additional X-ray optics) [65,101,102]. With appropriate X-ray optics (such as zone plates) and sources, even smaller spatial resolution down to 50 nm with a small field of view in the range of 10–20 μm is possible [103]. However, in both cases (geometrical magnification or by using additional X-ray optics), higher exposure time (due to low flux) and/or limited field of view are few associated drawbacks.(b)Synchrotron-based nanotomography. In synchrotron X-ray sources, the resolution can be enhanced at the detector level, for example by using a combination of thin scintillator detector with high magnification. With such set-ups about 200 nm pixel resolution can be achieved [104,105,106]. In addition, the size of the synchrotron X-ray beam can be further reduced by using X-ray optics such as a Fresnel zone plate, where the beam can be focused to a size as small as 50–60 nm diameter depending on the X-ray photon energy. Upon positioning the detector at a larger distance, higher spatial resolution as low as 30 nm can be achieved [105,107]. With the availability of coherent and high flux light sources at the 3rd generation synchrotrons the resolution of the radiography images can be further enhanced by coherent diffraction imaging (CDI)/ptychography [108,109,110].(c)Ptycho-tomography. Coherent X-ray beams have a constant phase shift i.e., waves are in-phase with each other. The transmission of the coherent beam through a material can be measured by placing the detector close to the sample. However, by placing the detector far away from the sample (e.g., >1 m), the transmitted wave interferes and produces a diffraction pattern image on the detector known as far field coherent diffraction. Such diffraction patterns can be used to retrieve the phase change of the propagated wave by an iterative phase retrieval algorithm [111]. When the diffraction pattern is measured across the entire sample, a spatially resolved change in amplitude and phase of the transmitted wave through the object can be reconstructed, and it is known as ptychography. An important feature of the ptychography technique is the ability to reconstruct computationally the phase and amplitude of the imaged object as well as the probe (i.e., the illumination of X-ray beam on the sample). The image reconstruction is carried out by different reconstruction algorithms [110,112,113,114,115,116,117], some of which have been implemented as open-source toolkits [118,119]. Since there are no optical elements involved in the image formation, the ptychography technique is theoretically diffraction-limited, and resolutions as good as 10 nm have been proven, also in 3D (as ptycho-tomography) [120]. However, measuring objects at such resolution is challenging and the field of view is limited, requiring special sample preparation in many cases. Due to the requirement of a coherent light source, ptychography is carried out using laser sources or at synchrotrons, however, Batey et al. [121] demonstrated the possibility of carrying out ptychography using lab-based X-ray sources. To the best of our knowledge, ptycho-tomography has not been used to characterize pharmaceutical solid dosage forms.Phase contrast tomography—Unlike conventional transmission radiography/imaging, where the reduction of the amplitude of the X-ray wave (intensity) is used to generate image contrast, in phase contrast imaging the phase shift induced by the object is retrieved. Pharmaceutical dosage forms are often made of organic compounds, therefore, different pharmaceutical compounds with similar elements can have a comparable attenuation coefficient making it difficult to identify individual ingredients (from absorption contrast) and can result in the need to add contrast agents or stains. To overcome this issue, phase contrast imaging can be an alternative [122,123]. At the X-ray energies (in the keV range) needed to image a full pharmaceutical tablet, the absorption component (β) is typically smaller than the refractive index decrement (δ). The latter makes the refractive index value different from unity, which results in the transmitted intensity to undergo a significant phase shift along with absorption [110,124,125]. While the amplitude of the transmitted image is a direct measurement of the intensity, the phase component is measured by modifying the measurement or by introducing additional optical elements on the X-ray path. The phase contrast can be measured by using interferometry methods [126], analyzer [122,127], and propagation based imaging, all with specific advantages and limitations [100,128,129]. An extension of the interferometry method is the grating-based differential phase contrast imaging achieved by using two different gratings in the optical path [130], and can be used for both non-coherent and polychromatic X-rays [131,132,133]. Zernike phase contrast imaging is another technique employed to measure phase contrast and can be implemented in lab-based tomography setups [103] as well as at synchrotrons [134,135]. Phase contrast can also be achieved at the detector level by edge illumination approach, where the X-ray passing around the edge of the sample is measured at the edge pixels of the detector [136]. By using this technique, the refracted beam is separated from the non-refracted beam, and the phase shift is analyzed. Holotomography is also an extension of phase contrast tomography, which can be carried out by exploiting the propagation-based phase contrast effect [124]. The reconstructed tomography images consist of spatially resolved refractive indices.Dark field imaging—The contrast in a standard radiography image represents the degree of absorption by the object. In a dark field image the contrast represents the degree of scattering from the object by filtering the non-scattered light, making it possible to identify sub-voxel resolution features. To achieve this, the light source is passed through certain optical elements (such as a dark field condenser lens, which is typically used for dark field imaging in optical microscopes), such that the non interacted beam can be filtered and only the interacted beam is measured. In X-ray microscopy, it is achieved by using a bright field stopper before the detector or grating interferometry [123,137,138,139]. The latter can also be used in lab-based polychromatic X-ray sources. Dark-field imaging is very complementary to attenuation and phase contrast, highlighting strongly scattering regions. As such, sub-voxel features can be visualized. Using tunable setups, specific feature sizes can be targeted [140]. To the best of our knowledge, dark field imaging/tomography has not been used for characterizing pharmaceutical drug products.Small angle X-ray scattering tensor tomography—Absorption based X-ray tomography is based on reconstructing radiography images with individual pixels representing the local absorption as a scalar quantity distributed across the sample. However, a tensor tomography consists of tensor field in each pixel, i.e., each point (voxel) in the sample is a multidimensional array (such as a 3 × 3 × 3 matrix) [141] which are then analyzed to obtain a 3D image with each voxel representing a unique vector quantity. In SAXS tensor tomography, the tensor field is the measure of local X-ray scattering determined by different rotation angles with respect to the X-ray propagation vector [142]. The measured scattering functions are then used to reconstruct the 3D tomographic image of the local reciprocal space and the structural orientation [143]. SAXS tensor field tomography is a relatively new technique and is suitable for samples which scatter less and the spatial resolution is dependent on the size of the X-ray beam used. Disadvantages of this technique include a long acquisition period and computationally intensive post processing time. The potential for pharmaceutical applications is yet to be explored, and it is particularly suitable to analyze the local crystallinity or the shape orientation of the individual particles in the solid dosage forms.Diffraction contrast tomography—The absorption or phase contrast tomographic images do not provide information on the crystallographic orientation. To measure the local crystalline structure, X-ray diffraction tomography or diffraction contrast tomography can be used, where the latter is similar to 3D X-ray diffraction microscopy [144]. Both techniques offer high sensitivity and spatial resolution upto 0.5 μm (at a synchrotron source), and can be achieved by using an appropriate detector such as a thin scintillator detector in combination with a charge-coupled device to obtain a magnified radiography image. The resolution is typically dependent on the beam size, type of detector, and the angular resolution (i.e., the rotation step size) [104,145,146,147,148] Pharmaceutical ingredients are often crystalline materials and the crystallinity can influence tablet characteristics such as solubility [56]. X-ray diffraction tomography measures the radial diffraction signal as a function of rotation angle and position on the sample. A global diffraction pattern of the sample can be obtained by integrating the entire stack of diffraction patterns, from which the necessary peaks are selected to produce the sinogram of a particular crystalline phase and carry out tomographic reconstruction. Similarly, other diffraction peaks can be selected and individual crystalline phases can be extracted. For diffraction contrast tomography, the radiography image consists of the absorption map as well as the diffraction spots upon satisfying Bragg’s law condition. The diffraction spots are separated from the radiography image, for example, by grey value thresholding and are analyzed based on the spatial and crystallographic criterion. Once the diffraction spots are analyzed, the local crystallographic orientation is calculated. In this approach along with the reconstruction, the data analysis is composed of many computational algorithms used to subtract the absorption component, analyse the scattering pattern and extract the different crystallographic phases [149,150,151]. Commercial systems to carry out laboratory based X-ray diffraction or diffraction contrast X-ray tomography are also available, where grain sizes down to 40 μm can be resolved [152,153].Spectral imaging—It combines spectroscopy and imaging techniques, such that one can spatially resolve the degree of absorption as a function of X-ray energy thereby identifying the local chemical states [154]. Such spatially resolved spectral/absorption images can be obtained in synchrotron sources by tuning the X-ray photon energies to the signature absorption edges of the material [155]. However, lab-based X-ray sources are polychromatic in nature with energy typically ranging from 1 keV–160 keV. As typical X-ray detectors only measure the total dose deposited on the scintillator material, the spectral information, i.e., absorbance signal of the different X-ray energies are mixed up, hence the chemical information is lost. To overcome these issues, spectral imaging can be applied by (1) using different source spectra or (2) by using spectral or photon-counting X-ray detectors. In the former, different (yet often overlapping) spectra are used, i.e., different energy ranges as in the case of lab-based dual microtomography setup, such that the ratio in the absorbance signal is different for different chemical components [156]. However, such methods typically have limited efficiency, and require good calibration. The same can also be achieved at the detector level. Spectral imaging detectors can be divided into multispectral and hyperspectral detectors. Multispectral detectors can measure photons with different energy ranges (or energy bins). They usually have relatively poor energy resolution and suffer from the charge sharing effect, yet promising results have been achieved to identify specific materials. Alternatively, hyperspectral detectors can be used, where the number of energy bins are higher, and can provide higher energy resolution than a multispectral detector. Nevertheless, the energy resolution achievable using a hyperspectral detector is still lower than what is achievable at synchrotrons where the photon energy is tuned by the monochromator, allowing for extremely high spectral resolutions (down to eV level at hard X-ray range). Implementing a hyperspectral detector system can have numerous challenges, and typically require large upgrades at the detector, such as with the electronics and data acquisition software [157,158,159,160,161]. Different (hyper)spectral detectors are being developed, and can be a suitable imaging tool for pharmaceutical compounds at high X-ray energies.

Many of the advanced tomography techniques have been developed and are only available at synchrotron facilities, and the implementation of such techniques at lab-based facilities is important for addressing broader applications. Nevertheless, conventional microtomography, which can be carried out both at synchrotron and at lab facilities, offers a strong added value in many research fields. A broad range of lab-based instrumentation is commercially available, and many lab-based and synchrotron facilities can be used by both academic and industry users.

The tomographic reconstruction of the 3D volume from the raw projection data is typically carried out by reconstruction software. Commercial licensed software are available, however, several lab-based instruments, and synchrotron facilities have custom made reconstruction software. Open source reconstruction software includes Reconstruction tool kit [162], STIR [163], Astra toolbox [164], TomoPy [165], TIGRE [166], Tofu [167], PyHST2, PyRaft. Additionally, some of the advanced techniques mentioned earlier (such as SAXS tensor tomography or diffraction contrast tomography) require dedicated pre-processing or post-processing steps, or even dedicated reconstruction algorithms. After reconstructing the 3D image of the sample, the data is typically produced as 2D image slices across the length of the object imaged perpendicular to the rotation axis, stacking all the images on top of each other yield the 3D tomography image.

Tomography images are rich in information, and analysis can be done on different levels.

Qualitative analysis—the datasets can be rendered in 3D for visual analysis, assessing, for example, the surface topology or the internal structure. The virtual volume can be manipulated using, for example, virtual cut-throughs, and the 3D spatial nature of the volume makes interpretation very intuitive. However, this interpretation is also the major limitation of such types of analysis.Quantitative analysis—the 3D volume can be analyzed by dedicated software, retrieving information such as pore/particle size distributions, density measurements, etc. Though such analysis methods can yield very interesting numerical results, they imply a data reduction, i.e., loss in information. An example hereof is the pore or particle size distribution, discarding the spatial information, hence neglecting areas with deviating pore sizes. Combining 3D analysis with 3D rendering can be a good way to overcome such limitations as shown in Figure 2.Modelling and simulation—Finally, the 3D dataset can be used as an input model for simulations, such as fluid flow simulations or finite element analysis. Such analysis based on real 3D data can be extremely powerful, but care must be taken in the extraction of the input model, and the researcher must be aware of the limitations of the input data.

3D visualization and analysis can be done by various commercial software such as Avizo (ThermoFisher Scientific Inc., Waltham, MA, USA), Dragonfly (Object Research Systems (ORS) Inc., Montreal, Canada), VGStudioMax (Volume Graphics GmbH, Heidelberg, Germany), Image Pro Analyzer 3D (Media Cybernetics, Inc., Rockville, MD, USA), Octopus Analysis [169], and open source software (with a graphical user interface) such as medical imaging interaction toolkit [170], paraview [171,172], 3D slicer [173], ImageJ/Fiji [174]. The data analysis software can be selected based on the user’s experience in programming, budget availability and the need for support. Data processing is a key to extract quantitative information from X-ray tomography images [175]. Several data processing functions are built-in within the visualization software. Few open source python modules that can be used to create image processing programs include PoreSpy [176], PyVista [177], python ODL, OpenCV. Often the data necessary to be extracted varies with samples, therefore, the built-in function cannot be used as such and requires additional data or image processing. While the type of image processing workflow varies, a general set of the data analysis begins with the segmentation of the required pixels/voxels from an image. The different types of segmentation methods are reviewed in Refs. [178,179]. A few of the segmentation techniques with potential for processing X-ray microtomography images include;

Grey value thresholding-which uses the grey scale of an image/histogram with either 1 or 2 boundaries to determine the voxels of interest.Object based segmentation-where segmentation is carried out by identifying groups of pixels by their shape or size (it is typically carried out by machine learning approaches such as training a convolutional neural network).Clustering based segmentation-where the pixel intensities are clustered by the algorithm into a number of groups based on input conditions.Iterative based segmentation-where the pixels are sorted based on mathematical models.

Once segmented, the pixels are grouped and labelled. Here the different pixels are individually identified, and the positions are recorded. This action can be carried out in 2D slices as well in 3D volume, the former is typically carried out for computational/memory reasons. Once the necessary data/pixels are extracted further analysis is performed, for example, the area or volume of the separated pixels, or other statistical analysis such as size distribution can be obtained. In addition to the conventional image processing techniques, newer machine learning techniques are also used to process the tomography images. In particular, for pharmaceutical tablets it is suitable for image segmentation [180,181,182], object detection such as pores [183], or classification of the data sets to identify different types of pharmaceutical tablets [184,185]. The important aspect of analysing the X-ray microtomography data of pharmaceutical tablets is how to couple the extracted information, particularly in the case of quantitative 3D analysis, to the functionality of the tablet or the characteristics of the excipients and the APIs, and link this to the production process in order to tune the manufacturing parameters. In the following sections, we describe in detail the tablet characterization listed in Table 1 and illustrate the usability of X-ray microtomography.

**Table 2 pharmaceuticals-16-00733-t002:** Summary of different modalities of X-ray tomography. F— with full field of view (e.g., tablet volume—1 × 0.5 × 0.5 cm^3^), PF–with partial field of view (e.g., 50 × 50 μm^2^). The measurement time corresponds to the time taken to collect multiple projection images to reconstruct a 3D image, and does not include the time taken for the tomography reconstruction and post processing. We indicate here a broad estimate of the measurement time, however, the measurements can also be made shorter or longer depending on the requirements such as the resolution or signal to noise ratio.

Modality	Resolution	Measurement Time	Applications
Lab-based microtomography	3–20 μm (F)	Minutes to tens of minutes (e.g., [186]). Faster acquisition possible, typically for 4D imaging (e.g., [187])	Determine density/content distribution, pores characterization, coating layer characterization, study of dynamic process
Lab-based nanotomography	0.3 μm–1 μm (PF)	Few hours to as high as 24 h, depending on the exposure time, resolution	Determine density/content distribution, pores characterization, coating layer characterization
Synchrotron-based microtomography	≥0.3 μm (F)	Seconds to minutes, depending on the field-of-view and resolution (e.g., [188])	For high resolution images, monochromatic X-ray can be tuned to characteristic absorption edge where applicable for chemical characterization, study of dynamic process (with higher time resolution but is limited by sample rotation)
Synchrotron-based nanotomography	≥30 nm (PF)	Tens of minutes to hours (e.g., [189])	For high resolution images, monochromatic X-ray can be tuned to characteristic absorption edge where applicable for chemical characterization
Ptycho-tomography	≥10 nm (PF)	Few hours (e.g., [190])	Imaging nanosize pores/particles, obtain pore network information, obtain simultaneous phase and amplitude information
Phase contrast microtomography (synchrotrons)	≥1 μm (F)	Measurement time similar to microtomography (e.g., [191]) (requires additional time for determining the appropriate position of the detector)	Enhance contrast for samples with similar attenuation coefficients
Dark field imaging (synchrotrons)	≥1 μm (F)	Tens of minutes to hours, depends on the source used, and desired quality and resolution, (e.g., [192])	Identify sub-voxel features below the resolution of the system and not visible in phase or absorption contrast imaging, enhanced contrast for samples with similar attenuation coefficients
Small angle X-ray scattering tensor tomography (synchrotrons)	≥50 nm (F)	Tens of hours (about 35 h for 25 μm voxel size [142])	Identify local orientation of the particles, crystallographic orientation, anisotropy
Synchrotron-based diffraction contrast tomography	≥0.5 μm (F)	Few hours (for diffraction contrast tomography [149])	Identify local crystal structure

## 3. Tablet Characterization

Once the mixtures/granules of APIs and excipients are converted into a tablet, different tests are carried out to optimize the production process and performance. The set of important parameters to check include mechanical strength, content distribution, crystallinity, particle morphology, (when needed) properties of the coating layer, as well as the dissolution process. The list of characterizations, the corresponding analytical approaches, and the equivalent X-ray tomography analysis are summarized in Table 1. The same can be applied to other types of tablets, such as freeze dried or 3D printed tablet (with some modifications) [193,194,195,196], and it will not be discussed here. We consider compressed tablets for their simplicity in the production and their wide usage in the pharmaceutical industry.

To obtain a compressed tablet the powdered or the granulated mixture is subjected to a compaction process to obtain the necessary tablet shape and size. Poor flowability, mixing or an uneven compressive pressure can result in an uneven distribution of powder/granule within the tablet such that certain regions of the tablet are denser than the other [51,197]. More importantly, the applied pressure affects the pore concentration and density which as a result affects dissolution, disintegration, clinical functionality and stability of the tablet. Compression of the tablet is typically carried out by die compaction. For uncoated tablets, compression is the final step in the tablet production. However, for coated tablets, after compression the tablets are subjected to a coating process. The coating is carried out for example by spray coating followed by a drying process that can typically induce mechanical and thermal stress on the tablet potentially affecting the compaction and integrity of the tablet [198,199]. Therefore, the compression step is a crucial stage in tablet manufacturing as all the other characteristics of the tablet can be affected by this step. During the compression process it is important to ensure that the compressed tablet is intact (i.e., solid and unbreakable), and stable from external mechanical stress until the administration. Therefore, the first step in the tablet characterization is the analysis of its mechanical strength.

### 3.1. Mechanical Testing

The mechanical properties of the tablet originates from the tablet design such as the structural design or the shape, and are often carried out by a theoretical approach or simulation such as statistical modelling [200,201], or finite element analysis [202,203,204]. While the theoretical values related to the mechanical strength can be calculated, the parameters after the tabletting are separately monitored in the production line due to the presence of possible manufacturing defects arising from friction [205]/sticking [206], thermo-mechanical effects induced by the compression instruments [207,208], contents [209] or by powder flowability [51,210,211]. A set of parameters measured to characterize the mechanical strength of the tablet include Young’s modulus, tensile strength and Brinell hardness [200,212]; and for compressed tablets it is measured as a function of compaction pressure, compaction speed and head flat type [197,203] (i.e., the surface of the punch) or other parameters such as the concentration of lubricants in the mixture. The Young’s modulus is the ratio of stress (force per unit area) and the strain (deformation), and is a measure of the tablet’s elasticity and its ability to withstand external forces. The tensile strength determines the force required to break the tablet and the Brinell hardness/indentation hardness is the measure of how the tablet deforms when pressure is applied on a particular spot on the tablet. While the type of tests varies, and other parameters such as strain-rate sensitivity, ductility or brittleness can also be measured [51,213,214,215,216]. The measurements are usually carried out using table top instruments or hand held devices [211,215,217,218,219]; and are typically destructive, therefore they are performed on few samples per batch. In certain cases a friability test is carried out i.e., to evaluate the behaviour of a tablet when subjected to collision, or pressure induced during the coating process. Possible non-destructive techniques for mechanical testing of tablets is reviewed by Dave et al. [220].

The aforementioned tests only give phenomenological information, and give limited to no information on the origin and why a certain effect happens. The microscopic origin of the mechanical properties is the pore concentration and density distribution within the tablet, and gaining knowledge on their relationship with the manufacturing process is of key importance in the tablet production and in the optimization thereof. Hence, characterizing the pores and density variation with spatial resolution using microscopy techniques such as X-ray microtomography is useful [221,222,223,224].

Some of the important reasons for the density variation or the pore formation in the tablet include the pressure variation on the punches, powder flowability and compactibility. This can be characterized by standard absorption-based X-ray microtomography where the local attenuation coefficient is calculated as demonstrated by Sinka et al. [186]. Figure 3a shows a representative reconstructed density distribution map of a single slice of a microcrystalline cellulose tablet where one can identify that sharp edges and the surface below the break line (notch) are more dense than the surfaces near the flat sides/edges. Density variation not only affects the overall tablet strength but can also influence the local mechanical strength on the tablet potentially affecting the dissolution process. Pores or microcracks on the other hand impact the ductility and break force of the tablet and are typically produced due to high compressive pressure. The concentration of pores is represented by porosity, which is the ratio of the volume of pores to the volume of the tablet. Pores are empty spaces, therefore have zero attenuation coefficient, making it easy to identify in X-ray microtomography images. While porosity can be described by a single quantifiable value (as ratio of volume of pores to the volume of tablet), representing the density variation is possible by visual images with the appropriate colour map of the density scale. However, the alternate approach to represent the density distribution is by plotting the density profile across the tablet slice. There are many ways to interpret the density distribution in microtomography images. For example, the density distribution map from X-ray microtomography can also be used to accurately estimate the compression pressure on the tablet by partial least square regression and finite element analysis as reported by Hattorii et al. [225]. The partial least square regression method can also be used to predict the content uniformity in the tablet from X-ray microtomography using density distribution data [225,226]. To simulate the compaction behaviour in the production line, dedicated compaction simulator instruments are used to replicate the production process [51,227,228]. Similar to the compaction simulator, dedicated devices are available for in situ mechanical characterization of materials in microtomography setups where images can be measured as a function of applied forces [67,68,229,230,231,232]. This approach can also be used to determine the dynamics of microcracks or the tablet breakage. Other non-destructive methods to characterize the density variation (with spatial resolution) include magnetic resonance imaging [72], acoustic methods [211,233], mapping of local hardness (indentation test - not completely non-destructive) [51,205] and autoradiography [234]. These techniques offer a lower spatial resolution when compared to X-ray microtomography. In addition to the non-destructive techniques, other sensing technology to monitor density variation as a process analytical technology has been reviewed by Stranzinger et al. [52]. Additional benefits to characterize density distribution include optimizing the powder flow parameters [235], processing temperature [236], and roller compaction parameters [237].

The porosity data can be extracted from the tomography images by image segmentation. For a data set with a good signal to noise ratio, the errors associated with the segmentation can be minimal, however, one can expect large errors in the segmentation with noisy data and samples with small pores in the length scale of the voxel size. Nevertheless, a simple approach to carry out segmentation in an image is by thresholding the grey values corresponding to the pores. Once the pores are segmented, the sum of the individual pixels multiplied by the voxel size is the volume of the pores in the tablet from which porosity can be obtained (volume of pores/volume of the tablet). Here, we emphasise that the origin of pore formation can also be from the granulation process [238], and such pores are referred to as intergranular pores which are often smaller than the pores or microcracks formed during the compression or between the filler (excipient) added to the tablet [239]. One can consider both microcracks and intergranular pores are related to the mechanical strength of the tablet. High porosity can result in low tablet strength as the tablets break from the pore region. The effect of compaction pressure, porosity and Brinell hardness is shown by Busignies et al. [212]. Typically, the Brinell hardness changes inversely with the porosity of the tablet, and the porosity reduces with increase in compaction pressure, which in turn can increase the mechanical strength of the tablet [e.g., Figure 3b].

Porosity is therefore an important parameter to represent the strength of a tablet, as it is proportional to the brittleness, i.e., how easily the tablet can break [240,241]. Furthermore, the pores provide the necessary surface area for the fluid to penetrate and in turn influence the tablet disintegration, drug release/bioavailability and dissolution [242,243,244,245,246]. Hence, in certain cases the presence of porosity can be a necessity, and it is important to understand the role of pores beyond its contribution towards the mechanical strength. The pore size can vary between nm to μm length scale, and with lab-based X-ray microtomography set-ups the resolution can be restricted up to a μm. Hence, destructive techniques such as SEM or atomic force microscopy (AFM) need to be used to evaluate the smaller pores. Nevertheless, porosity can also be measured by non-microscopy methods without spatial resolution such as by He pycnometry or mercury porosimetry [247]. The porosity value links the macroscopic mechanical strength of the whole tablet, density variation can provide information on the local mechanical strength.

In summary, the microscopic origin of the mechanical strength of a tablet is related to the density variation and the porosity distribution. X-ray microtomography can be a suitable tool which can provide information on these two parameters. The detection of individual pores depends strongly on the achievable resolution of the microtomography setup, which can result in a lower estimation of the net porosity of the tablet if smaller pores exist. Nevertheless, previous studies have shown that such an approximation can be sufficient to estimate the macroscopic mechanical properties as well as the tablet dissolution properties as these larger pores often have the strongest influence on these properties. Additionally, although smaller pores cannot be visualized directly, their presence alters the local density of the material, enabling to study variations in porosity within the sample. The density variation provides information on the spread of the compaction of the tablet, however, information on the spread of the pharmaceutical ingredients is still missing, as a standard X-ray microtomography characterization is not element specific. When a tablet consists of multiple pharmaceutical ingredients, additional measurements or data analysis are required and is discussed in the next section.

### 3.2. Content Distribution

In a tablet, in addition to the APIs, one or many more excipients are added. These excipients can be; diluent (to enhance tablet properties, such as improved cohesion or to promote flow), surfactants (to lower surface tension and enhance disintegration), fillers (to size up the tablet to the required size), binders (to induce granule and compact formation), disintegrants (to break down the tablet in the fluids of the body), wetting agents (to assist with the dissolution of hydrophobic active pharmaceutical ingredients), lubricants (to reduce the friction between powder, punch and die), anti-adherents (to prevent sticking during the production), glidants (to enhance powder flow during tableting) and anti-oxidants/metal chelating agents (to help stabilise chemically the active pharmaceutical ingredients) [51]. With multiple ingredients used in the tablet formulation, often a granulation procedure is carried out [248]. Granulation is helpful in different ways, for example, it helps in better binding of ingredients if the powder compactibility is less and provide better flowability in the production process [249]. The ingredients are blend together before the granulation process. However, during the granulation process, the ingredients in the blend are subjected to physicochemical changes, and can potentially redistribute the contents such that the blend is no longer uniform within the granule [186,250,251]. In addition, the same can occur during the transport or the compaction process where the ingredients move due to the friction or uneven applied compressive pressure, respectively. An uneven compressive pressure can also result in uneven spread of the concentration of APIs and excipients in the tablet. A non-uniformity of the ingredients would result in different dosage of the therapeutic drugs [252] or dissolution process [253,254,255]. Hence, it is important for the quality of the drug product to identify the different ingredients (also impurities) and determine whether the content homogeneity is maintained through out the manufacturing process [256,257].

Typically, X-rays can be used to characterize and identify the chemical state of the sample either by spectroscopy or spatially resolved spectro-microscopy methods [89]. For the latter the X-ray photon energy is tuned to the signature absorption edges to acquire a radiography image. However, obtaining chemically resolved absorption images from a large sample such as a pharmaceutical tablet using high energy X-rays (i.e., >1 keV) in X-ray microtomography is challenging. In lab-based X-ray microtomography systems, with high energy (>1 keV) and polychromatic X-ray sources, tuning the X-ray energy to specific absorption edges is not possible to obtain a chemically resolved absorption images. However, by using two different X-ray sources or one X-ray source with two different ranges of X-ray energy (spectrum) configurations, the chemical resolution can be partially enhanced [258]. To the best of our knowledge, this has not been applied for pharmaceutical dosage forms, and is typically applied in (pre-)clinical imaging using contrast agents [258,259]. In synchrotrons, despite being able to use a monochromatic beam (typical bandwidths 1 × 10${}^{-2}$ or 1 × 10${}^{-4}$ eV), the typical energy range used for X-ray microtomography experiments is between 4 keV to 50 keV (for its ability to use the beam under atmospheric pressure and longer propagation distance) [106,260,261,262,263,264,265,266,267,268,269,270,271,272]. Therefore, only the absorption edges falling within this energy range can be chemically resolved [273]. The absorption edges of organic compounds with C, N, O, H which are often used in pharmaceutical compounds do not fall in this energy range and have very subtle differences in the attenuation coefficient with different organic compounds. The energy range to characterize the organic materials typically lies in the soft X-ray range in what is known as the water window with energies ranging from 230 eV to 540 eV (between carbon and oxygen k-edge). To differentiate organic compounds merely tuning the X-ray photons to the characteristic absorption edges of C, H or O is not sufficient, therefore the X-ray photons need to be tuned to near absorption energies which is typically few meV above or below the characteristic elemental absorption edges [274,275]. Such characterization is referred to as near-absorption edge spectroscopy or microscopy. Nevertheless, with soft X-ray energies a non-destructive characterization of a pharmaceutical tablet is not possible due to the low attenuation length (in the order of nm to μm range) when compared to the size of the tablet (in cm). Therefore, obtaining a perfect chemically resolved X-ray tomography image is difficult in both lab-based and synchrotron-based X-ray tomography setups. To overcome this issue, imaging by phase contrast X-ray tomography or with a hyperspectral detector can be an alternative, and the later it is yet to be investigated for pharmaceutical tablets. Figure 4a,b shows an example of a microtomography image slice of a tablet from Janssen Pharmaceutica measured at the lab-based X-ray microtomography setup at the the Ghent University, Center for X-ray Tomography (www.ugct.ugent.be, accessed on 1 April 2023) [65], and by phase contrast tomography at Anatomix beamline, Synchrotron SOLEIL, respectively. The tablet is composed of nine different components (the contents of the tablet are confidential) and Figure 4c,d shows the grey value distribution of Figure 4a,b respectively. From the images and the histogram one can identify that the contrast difference with the phase contrast synchrotron microtomography image obtained at 40 keV is higher than the lab-based microtomography image with X-ray energies ranging from 10 keV to 100 keV. While certain phases indicating different components, can be identified or grouped from the phase contrast X-ray tomography image, the chemical resolution is still low. Nevertheless, particle distribution by X-ray tomography can still be carried out on tablets where various ingredients can be identified and segmented.

A quantity used to describe spatial distribution (i.e., uniformity of pharmaceutical ingredients within a compressed tablet) is “homogenity” or “mixing homogenity” which can be used to quantify the chemically resolved images. The analysis (algorithm/guideline) was developed by Rosas et al. [276,277,278], in the frame work of process analytical technology initiative of the US Food and Drug Administration (US FDA). Another quantifier of the spatial ingredient distribution can be described by Distributional Homogeneity Index developed by Sacré et al. [279]. The two quantifiers use different approaches, while having many attributes in common. The common procedure involves segmenting the ingredients by *k*-means clustering, labelling the individual particles and finally splitting the large size 2D image into smaller macropixels [280] (i.e., an image with 100 × 100 pixels is split into 100 − (10 × 10 pixel images) to carry out the statistics. The macropixellation provide higher spatial resolution towards the homogeneity. The choice of the size/number of macropixels is illustrated by Hamad et al. [280]. Within each macropixel statistical analysis is carried out. The quantifiers to determine the extent of mixing includes Lacey index [281], Poole’s index and segregation index [276,282]. The distributional homogeneity index uses a least square regression model to determine the distribution map which is later compared with the randomised image to obtain the homogeneity index. Nevertheless, these quantifiers were developed for 2D NIR or Raman microscopy images, where individual pharmaceutical ingredients can be identified. A 3D macropixellation approach to calculate the homogeniety of a tablet in a 3D volume measured by microtomography is illustrated by Schomberg et al. [283]. Figure 5 shows an example on how a tablet volume can be split into multiple 3D macropixel volumes. In each 3D macropixel, we measure the extent of uniformity of two components present in the tablet. A red color indicates a high level of uniformity with equal concentrations of both components, while blue indicates a non-uniform region.

The spatial distribution of the ingredients in X-ray tomography images is typically quantified as the volume fraction of the pharmaceutical ingredients to the total volume of the tablet. Figure 4e shows a representative example of ingredient distribution in a moxidectin tablet reported by Wagner-Hattler et al. [188], where the volume fraction of the pharmaceutical ingredients of the entire tablet at different radial distances from the center is extracted. While the segmentation of ingredients on tablets prepared by powder compaction remains a challenge, tablets consisting of structurally distinguishable ingredients such as circular shaped granules are easier to identify and segment in the images than random irregular shaped particles. Dale et al. [284] demonstrated an efficient segmentation by grey value thresholding and represented the distribution by volume fraction of different ingredients as a function of radial distance from the center. Other granule type ingredient distribution using X-ray tomography has been reported by Liu et al. [285], Csobán et al. [286] and Zhang et al. [287]. Due to the low chemical resolution in conventional X-ray microtomography, substitute characterization such as SEM/EDS or Raman microscopy with chemical resolution is carried out separately or in combination with X-ray microtomography.

In summary, chemical identification of pharmaceutical ingredients by a non-destructive approach is an important analysis during the developmental stage. For more accurate X-ray based measurements, one should consider using soft X-ray microscopy approaches, however, it is destructive. Nevertheless, with a microtomography setup using hard X-rays, one can consider phase contrast mode, by which certain contrasts can be enhanced and supplemented by SEM or Raman microscopy. Once the necessary pharmaceutical ingredients are identified in the X-ray microtomography images, further analysis such as on spatial distribution, or homogeniety can be carried out by 3D macropixellation approach. Certain pharmaceutical ingredients have distinct particle shapes or sizes that can be identified or segmented during image analysis. Particularly, analyzing the shape or the size of pores can provide information towards the dissolution characteristics. Similarly, the pharmaceutical ingredients can also be crystalline in nature and may have distinctive features. Methods to characterize these intrinsic properties such as shape, size or crystallinity are discussed in the following section.

### 3.3. Intrinsic Properties

With the availability of high-resolution microscopy techniques, one can characterize the physical properties of individual particles or pores in the solid dosage form in its powder as well as in tablet form. Intrinsic properties of pores or particles (also individual granules) typically include network properties, frequency distribution of their shape, size or crystallinity. For example, quantities such as the connectivity of the pore network provide information on the extent of dissolution or tablet disintegration. Linking such characteristics to the bulk characteristics is essential in tablet manufacturing, and was made possible with high resolution 3D imaging such as X-ray micro/nano-tomography. In the image analysis workflow, after the segmentation, the individual particles (or features of interest) are identified, and this can be achieved by using the contour algorithm or watershed algorithm, for example. Once the particles are identified, they are labelled and then the necessary information on their physical properties such as diameter, surface area or elongation and the corresponding statistics are extracted. The same approach can be carried out for the pore size distribution. Such analysis is possible in most of the image analysis software (e.g., Avizo (ThermoFischer Scientific Inc., Waltham, MA, USA) or ImageJ with MorphoLibJ plugin [288]). However, they typically discard the spatial information, i.e., the information on which region might deviate from the global characteristics will be missing and it is more suitable for visualization. Hence, the representation of these values through macropixellation, as described in the previous section can be useful. This is particularly suitable for example to identify pores size distribution across the tablet. Nevertheless, the individual features extracted can be further analyzed for its shape, anisotropy, tortuosity, permeability or connectivity where the latter three quantities are relevant for pore characterization. The shape of the particle/pore is classified based on the elongation, flatness and sphericity and can be applied to 3D volume [289,290]. The anisotropy refers to the orientation of the particle which is measured from the eigen vector which is deduced from the length of the particle in all directions [291]. Understanding the pore network through its tortuosity, permeability and connectivity provide information about the fluid flow and the dissolution process. These results can be used as input parameters to simulate quantities such as water transport [292,293]. Tortuosity is a measure of how straight or twisted is a pore structure, essentially the ratio of the distance the fluid must flow to the displacement between the opening and end of the pore. A detailed review on these analysis along with experimental examples can be found in Markl et al. [243]. Such detailed analysis on the individual pharmaceutical components using X-ray microtomography is demonstrated in Refs. [287,294,295,296,297,298,299,300,301]. While performing these analysis is useful to obtain the shape and size of the particles, one can expect a high degree of error especially with noisy images, low resolution images, irregular and complex shaped pores/particles. Therefore, with limited spatial resolution parameters such as smoothness or roundness can be difficult to measure. Hence, in addition to the computational data analysis, visual inspection is required [295,300]. Figure 6a shows a representative example of a segmented tomography image of a synthesised polymorphic clopidogrel bisulphate particles from the report of Yin et al. [297]. Here, the individual particles are separately assessed, labelled by their volume and were given separate colour codes. Once labelled the necessary voxels of the individual particles (in this case clopidogrel bisulphate) are extracted and are shown separately in Figure 6b in red. This is particularly useful for characterizing the particle’s morphology to identify two different clopidogrel bisulphate crystallography phases. From the separated voxels the shape parameters such as volume, sphericity or surface area are extracted and can be plotted as a histogram. An example is shown in Figure 6c illustrating the volume distribution of two different crystal phases of clopidogrel bisulphate particles investigated in the study. The data analysis here is carried out using VGStudio Max (Volume Graphics GmbH, Heidelberg, Germany) and Image Pro Analyzer 3D (Media Cybernetics, Inc., Rockville, MD, USA). Similar characterizations on individual pharmaceutical ingredients using X-ray microtomography can be found in Refs. [294,295,297,301,302,303].

The synthesised pharmaceutical compounds are either monocrystalline or polymorphic in nature. For example, a tablet consisting of a single monocrystalline API will have uniform physical and chemical properties, such as uniform dissolution. However, the pharmaceutical ingredients are often polymorphic in nature. Polymorphism can also be induced during the compression, moisture or heat resulting in a change in the crystallinity/crystalline phases braking into amorphous states [304,305], or vice versa [306,307,308,309]. The degree of polymorphism has a strong effect on the dissolution rate, absorption rate, efficacy and bioavailability of the pharmaceutical compounds. One of the challenges in the production of crystalline pharmaceutical dosage forms is the requirement to retain the crystalline state during the production (e.g., during granulation, drying or compaction) [56,57]. While the crystallinity is characterized in the preformulation stage in the powder form or by simulation [310], to characterize the crystallinity after the tableting with spatial resolution can be challenging. Two-dimensional X-ray diffraction which can be carried out in a lab-based X-ray diffractometer set-up can be one possibility. In a 2-D X-ray diffractometer a focussed X-ray beam is pointed at a certain point on the sample and the diffracted beam is measured as a function of different rotating angles. Such characterization has been demonstrated by Thakral et al. [209,306], Koranne et al. [311], where the diffraction signal was obtained from one of the axis of the tablet (across the length) from which the crystallinity fraction was calculated across the length of the tablet. The crystallinity fraction is the ratio of the volume of the sample with diffraction pattern observed due to Bragg’s condition to the volume which are amorphous with no diffraction signal observed. Figure 7 shows the example plot of the crystallinity fraction as a function of the length from one of the axis of amorphous indomethacin tablets with different compressive pressure characterized by Thakral et al. [306]. Similarly, Koradia et al. [312] carried out crystallinity characterization as a function of tablet depth using grazing incidence X-ray diffraction, where the penetration depth of the X-ray was controlled by the incidence angle. With 2-D diffractometer characterization, one can identify crystalline and amorphous regions. Extracting the local crystallography information can be challenging due to its low spatial resolution, i.e., if the individual crystal grain is smaller than the size of the X-ray beam. The equivalent X-ray tomography approach to characterize the crystallinity in the sample is by X-ray diffraction or diffraction contrast X-ray tomography. To the best of our knowledge, such diffraction-based X-ray tomography on pharmaceutical tablets for non-destructive analysis has not been published. Other non-destructive approaches such as NIR spectroscopy [313], Raman spectroscopy [314], NMR/FTIR [315], has also been used to study the crystallinity in the tablets and an extended review of different characterization methods can be found in Brittain [316].

In summary, the intrinsic parameters of a particle such as shape, size or crystallinity could be used to identify the components that cannot be chemically identified through the absorption value in X-ray microtomography data. Upon identification, analysis on its distribution and its statistics can be extracted. For a better estimate on the local crystallinity, diffraction contrast tomography or small angle X-ray scattering approaches need to be considered. The intrinsic characteristics related to pores include anisotropy, tortuosity, permeability, shape (spherical, flatness etc.), and pore network properties. These parameters are particularly useful for tablets where the individual pores are interconnected. Disconnected pores should be analysed individually and the corresponding tortuosity or permiability will also be localized. Analysing these parameters will be suitable for controlled drug release tablets, 3D printed tablets and tablets with long dissolution rate.

### 3.4. Coating Thickness Analysis

A compressed tablet may be coated with a layer for different reasons, such as protection, taste masking or controlled drug release. The type and properties (thickness, permeability, and roughness) vary with the type of tablet, functionality and type of administration [317]. Typically, the role of a coating layer is towards a controlled drug release and enhanced bioavailability [247,318,319]. Critical parameter of the coating layer is the coating thickness uniformity and their distribution. The thickness of the coating layer can typically vary from 5 to 100 μm [198], and the uniformity of the coating layer is affected during the coating process [199,320,321]. The performance and the quality of the coating layer can also be determined by simulation and modelling tools such as discrete element method, and can be used to optimize the coating process [322]. The tablet coating involves spraying of the coating material in the liquid form by constantly mixing/rotating the container with the tablets, followed by a drying process [198,199]. Possible sources of defects include the formation of pores, cracks or clustering of the materials at a particular region [198]. The coating process is carried out by pan coaters or fluidized bed systems and possible newer approaches such as injection molding [198,323,324]. Typical coating materials include a mixture of ingredients such as cellulose, polymers or sugar, depending on the required functionality [325]. On an industrial scale, the coating process is monitored by characterizing the weight gain of the tablets, and the uniformity is characterized on the batches by SEM and finally by dissolution [326,327]. The coating layer thickness can be characterized by different non-destructive approaches such as X-ray diffraction [50], NIR [328], or Raman spectroscopy [329], which is confined to a smaller region with no spatial resolution. However, a combined spatial resolution and larger field of view is possible by optical coherence tomography [86,87], THz tomography [85,330,331], and X-ray microtomography [332,333].

For X-ray microtomography the coating thickness can be calculated from tomography images as long as the coating layer can be segmented and that the thickness is sufficiently large compared to the voxel size (>1 voxel size) to carry out quantitative analysis. In addition to the coating thickness, the presence of pores in the coating layer can be characterized by X-ray microtomography due its high resolution. After the coating layer segmentation, the thickness of the coating layer in 3D volume or in a 2D slice is typically obtained by using Ray method or Sphere method [334,335]. The sphere function is widely used and is available in open source python package such as in PoreSpy or ImageJ with MorphoLibJ plugin [176,288], and is also implemented in commercial 3D analysis software packages. A representative example of coating thickness analysis obtained from X-ray tomography by Ariyasu et al. [328] using VGStudioMax (VolumeGraphics GmbH, Heidelberg, Germany) is shown in Figure 8a, where the colour in the scale bar indicates the local thickness. In addition to a 3D visualization of the local thickness, the thickness distribution can also be calculated (i.e., thickness vs. frequency of occurrence) on different sides of the tablet to verify the uniformity [296,328]. The Ray and Sphere method may not be accurate to measure thickness for complex surfaces, and in such cases these approaches are only suitable for qualitative analysis or visualization. For complex coating surfaces, an alternative approach to measure the thickness with spatial information is to measure the radial thickness at different positions on the coating layer, and has been demonstrated by F. Sondej [332,333]. A representative image is shown in Figure 8b, the thickness of the coating is measured in a radial axis from the center where the spatial resolution of the thickness may depend on the angular step size of the radial axis. While the thickness uniformity can be measured by X-ray microtomography, measuring the surface roughness across the tablet surface requires further data processing. Such analysis has been demonstrated [see example in Figure 8c] using images from optical coherence tomography [247], where the roughness profile was extracted from the images and statistical roughness root mean square was calculated. Analysing the surface roughness is useful to investigate the tablet swelling or erosion during the dissolution process. Calculating the roughness from the microscopy images is often limited to the pixel size resolution, therefore, for higher spatial resolution i.e., in nm scale, measurement technique such as AFM has to be used. The coating layer can play a vital role in a controlled drug release process, since it is the first point of contact with a dissolution medium. With high resolution X-ray microtomography, the internal structure of the coating layer such as the presence of pores, or cracks can be identified, as they can alter the erosion/swelling rate of the coating and in turn the tablet dissolution. Therefore, X-ray microtomography can be a suitable technique to study the 3D thickness distribution as well as the coating internal structure/or its integrity and the roughness.

### 3.5. Dissolution Analysis

The dissolution of the tablet and the drug released after the administration is the vital step towards its therapeutic effect. Dissolution is the measure of the extent and/or the rate at which the drug is released in a solution (for example fluid in the stomach). Therefore, the dissolution characteristics or the drug release of the tablet determines the bioavailability of the pharmaceutical ingredient and in turn the dosage of the drug to be administered [247,318,319,336]. The dissolution rate is affected by many factors, such as the coating layer, density distribution, ingredient properties and pore concentration which are discussed in the previous sections. Furthermore, it is an important property for the final tablet quality and regulatory approval [14,15,18]. For a tablet with coating, the first point of contact of the tablet with the fluid is the coating layer. The dissolution of the coating layer occurs by erosion of the layer or by swelling. The erosion process results in layer by layer removal of the coating, while swelling of a tablet involves absorption of the fluid by the coating layer, which is common for polymer coating. The swelling mechanism results in an increase of volume of the coating layer up to a certain threshold volume after which the coating layer breaks to release the drug [337,338]. The release rate can also be effectively controlled over longer periods by using polymer excipients [339,340], delivery devices/systems such as multi-pellet formulations [341,342], membrane-controlled systems or osmotic systems [343,344]. Once the coating layer is dissolved, the dissolution rate is dependent on the intrinsic properties of the pharmaceutical ingredients. For tablets without a coating layer the dissolution is only dependent on intrinsic properties. The particles in contact with the fluid starts to dissolve. At the same time the fluid enters the tablet through the pores and its network resulting in the disintegration of the tablet into smaller pieces and this stage is effectively controlled by the concentration of pores [242,243,244,245,246,293,345,346,347]. By extracting the information of the pore network and its permeability from a SEM image, Zhang et al. [348] has shown that one can estimate the drug release rate by computational fluid dynamic simulation. Similarly, microtomography images are used to validate computational dissolution models such as in silico tools [349]. Typically, the disintegration time of the tablet varies exponentially with the tablet strength and density [51]. Adding disintegrants to the tablet will also assist the tablet to disintegrate. The disintegrants will swell due to the absorption of water and break up the tablet.

To evaluate the drug release rate in vitro, lab-based dissolution test is carried out using a dissolution bath setup [58,350]. In this setup, the release of API in the dissolution medium is monitored by high performance liquid chromatography or UV spectroscopy as a function of time. While this method is widely used and approved by regulatory agencies, it lacks 3D spatial information of the disintegration process and its mechanism during the dissolution, as the UV spectroscopy measures only the mass fraction of the compound in the solution. Such information can be useful for drug product development and understanding the drug release mechanism. Characterizing a dynamic event such as the dissolution process in real time with 3D spatial resolution is very challenging in X-ray microtomography, also in other 3D imaging and tomography techniques. The movement of objects during the image acquisition would result in poor reconstruction of the tomography images. Nevertheless, quasi-dynamic measurements can still be performed for a tablet with a slow disintegration rate, and that the movements of the particles during the disintegration are slower than the image acquisition time.

Characterizing the dissolution process with spatial resolution is typically carried out by MRI or X-ray microtomography. The former have both spatial and in some cases better chemical resolution than X-ray microtomography, and is widely used to characterize in situ tablet disintegration [44,45,351,352,353,354,355]. Other alternative techniques to characterize the disintegration/dissolution process include optical microscopy [356], UV-Vis microscopy [357,358,359], Raman based microscopy [255,360,361,362,363,364], and FTIR microscopy [365], however, the characterization is usually limited to a cross section or surface of the tablet with long dissolution time. In MRI, a 2D image of an area of 80 × 80 μm^2^ with section thickness of 600 μm can be acquired in 75 ms [354]. For lab-based X-ray microtomography, by using appropriate detector, 2D radiography images of much larger area (1–2 cm^2^) can be acquired in a time scale of less than 50 ms. The faster acquisition is possible because of the use of a 2D array detector, while MRI requires a raster type scan to spatially resolve the response signal of the applied oscillating magnetic field. In cases where dissolution time is very short (less than the image acquisition time for 3D reconstruction) 2D radiography images of the dissolution process can be acquired, and quantitative analysis on the area of swelling, or dissolution can be carried out [191,366,367]. While quantifying the area expanded in the radiography image is a good approximation to quantify tablet swelling, Laity and Camer [368] demonstrated the use of embedded glass microsphere tracers to track the dynamics of the swelling. The radiography images tracking the movement of the tracer during the swelling are shown in Figure 9a. This method provides information on the different swelling sites and the local rate of swelling. Acquiring sequences of 2D radiography images can provide uninterrupted information of the dissolution process with better time resolution. The drawback of using 2D radiography images is the lack of 3D spatial information.

Tablets with a very slow dissolution rate or tablets that can be temporarily removed from the dissolution process, can still be characterized to obtain a 3D image, and such tablets are typically characterized using X-ray microtomography. The protocol to carry out time resolved dissolution for such tablets involves immersing the tablet for a finite period of time followed by drying the tablet to remove the fluid from the tablet. The presence of fluid can further cause the tablet to dissolve causing objects to move during image acquisition, therefore the tablet has to be completely dried before X-ray microtomography images can be acquired. The drying process can be carried out either by oven drying, freeze drying, paper drying, or sol gel absorbance (or a combination of these). The process is repeated at different time points to characterize the evolution of the dissolution process. Here, only the leftover regions are imaged, with no information on the particles which got disintegrated or dissolved. The tablets characterized to study the evolution usually have a longer dissolution time, and the time resolution associated with these measurements typically ranges between 30 min to over 1 h [168,191,298,366,369,370,371,372,373,374,375]. The dynamic events which are of interest are tablet disintegration, swelling and the mechanism of the dissolution process.

Figure 9b shows 3D phase contrast X-ray microtomography images of the evolution of chitosan-λ-carrageenan matrix-based tablet swelling as reported by Yin et al. [191]. Here the tablets were immersed in a dissolution fluid for a fixed duration and dried in an oven at 60 ${}^{{}^{\circ}}$C for 2 days. The tablet is imaged after 1 h, 4 h and 12 h of the dissolution process. A 3D reconstruction was useful to identify various regions of the tablet, such as the evolution of the cross links in the polymer matrix shown in Figure 9c. Furthermore, for tablets where the dissolution is very slow (in the order of several hours), that is longer than the image acquisition time, X-ray microtomography can be carried out using a flow cell apparatus, where the time resolved 3D tomography images can be acquired without drying the tablet [376,377]. Figure 10 shows an example of such setup used by Moazami Goudarzi et al. [187] to characterize the dissolution of 3D printed tablets using a gantry rotating microtomography scanner [378]. In addition to the need of a fluid medium to create the drug release from the tablet, an alternative can be a high intensity focused ultra sound, which can be used to induce movement on the pharmaceutical ingredients mimicking the drug release process in the tablet as demonstrated by Fei et al. [379]. However this approach may not be suitable to replicate a real dissolution process. The dissolution data obtained from X-ray microtomography are typically analyzed visually, and are quantified by the change in volume, particle size distribution and porosity. The erosion is characterized by radiography images by analyzing the area of the tablet expanded and with the presence of the tracer, the position or the displacement is tracked and plotted as a function of time. One can also quantify the rate of dissolution by analyzing the fractal dimension of the tablet, which is the measure of the higher dimensional change or a measure to describe complex structure. Example of which can be found in Quodbach et al. [355] and Yin et al. [380] Similarly, other suitable parameters to quantify the dissolution rate from microtomography image include percentage object volume (ratio of dissolved volume of a particular phase to the total volume of the tablet), structure thickness (thickness of a particular phase/particle), structure separation (distance between two particles/phases), fragmentation index (change in the connectivity between particles/phases) and object surface volume ratio [381]. A review on the techniques used to characterize the tablet swelling and erosion can be found in Huanbutta et al. [382].

While the existing X-ray tomography setups are helpful to characterize different aspects of the tablet in a dissolution process, having a 4D resolution (3D characterization with time resolution) could potentially enhance the characterization of the dissolution process with in-depth details of the particle dissolution mechanism. Currently, the dynamic X-ray microtomography is being developed, and is available on few synchrotron beamlines with sub second time resolution. It is achieved by fast rotation of the sample and image acquisition in the detector [383]. In such experiments the samples are typically rotated in the rate of 50–1000 revolutions per second and has been carried out to study heat induced changes on metals. Such an approach may again not be suitable for mobile sample environments such as the dissolution process of a tablet due to the impact of the centrifugal forces on the sample during the rotation. Slow rotation of the sample is not suitable if the dissolution process is faster which results in a loss of temporal resolution. Replacing the dissolution fluid with a highly viscous fluid results in slowing down the dissolution process and can be a possible alternative, however this approach may not replicate the actual dissolution process. Nevertheless, set ups with a rotating gantry such as the Environmental micro-CT (EMCT) scanner at UGCT [187,378], or commercially available DynaTOM scanner from TESCAN, can be a possible alternative to characterize dynamic events, where the sample remains stationary while the source and the detector rotates around the sample to obtain the projection images. However, being based on X-ray tubes, the X-ray flux is much smaller, resulting in longer acquisition times as compared with synchrotron setups. Furthermore, advanced reconstruction algorithms by which reconstruction artifacts due to fast data acquisition, or the ability to carry out reconstruction with a low number of projection images (such that the acquisition time is lower than the rate of dissolution) are being developed [384].

## 4. Conclusions and Future Outlook

With a wide variety of drugs produced in the pharmaceutical industry, comes variety of research question and the requirement of different equipment or techniques. This hinders the presence of a uniform experimental and data analysis workflow for X-ray microtomography characterization in the pharmaceutical industry, resulting in a continuous adaptation of experimental protocols. Nevertheless, in this overview, we described the important list of characterizations in producing a pharmaceutical tablet, and illustrated the possibilities to use X-ray microtomography as an alternative technique to standard methods used in the pharmaceutical industry. From a single tomography measurement, information on the pores, particle morphology and density distribution can be obtained from which the mass distribution, mechanical and intrinsic properties can be analyzed, supporting the robustness of the technique. The content uniformity or distribution, as well as the coating layer, can be evaluated depending on the type of tablet and its ingredients. A non-destructive characterization of the pharmaceutical tablets to test the quality and performance is an important aspect of tablet manufacturing. Different techniques including X-ray tomography is being explored along with suitable measurement and data analysis approaches as described in this review. While X-ray microtomography has been largely used to characterize the solid dosage forms, it can also be used for characterizing other dosage forms such as creams, gels or suspensions, where sedimentation like issues can be addressed [385].

From the measurement aspects, phase contrast or dark field tomography can potentially offer higher benefits towards the characterization of pharmaceutical tablets and its compounds as it offers better contrast between compounds with similar attenuation length. With the availability of computer clusters large data sets can be analyzed, and complex computations can be carried out in a faster time scale. The developments on experimental capabilities or computational algorithms are needed to boost the possibility to carry out dynamic 3D imaging, which can bring a positive impact not only for pharmaceutical science but also for various other fields in science and technology.

Furthermore, with the availability of newer tomography techniques, properties such as crystallinity can be characterized by X-ray diffraction tomography. When needed high resolution images can be obtained by ptycho-tomography where pores or particle morphology can be characterized with sub 10 nm resolution on a small field of view. Still many X-ray tomography techniques remain unexplored and are yet to be used in pharmaceutical science. Nevertheless, the proof of concept experiments have shown that X-ray tomography and its modalities can be used to address various problems in pharmaceutical science, and continue to demonstrate that X-ray tomography can be an excellent and indispensable technique for non-destructive analysis of pharmaceutical materials.

## Figures and Tables

**Figure 1 pharmaceuticals-16-00733-f001:**
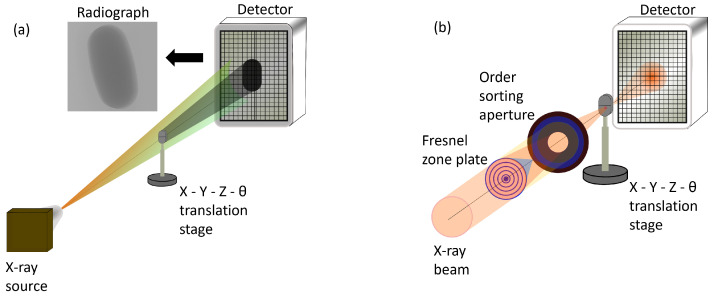
(**a**) Schematic of lab-based microtomography setup with cone beam shaped X-ray. (**b**) Schematic of synchrotron-based tomography setup with parallel beam configuration. The additional optics such as the Fresnel zone plate and order sorting aperture are needed for more advanced tomography techniques such as nanotomography/ptycho-tomography.

**Figure 2 pharmaceuticals-16-00733-f002:**
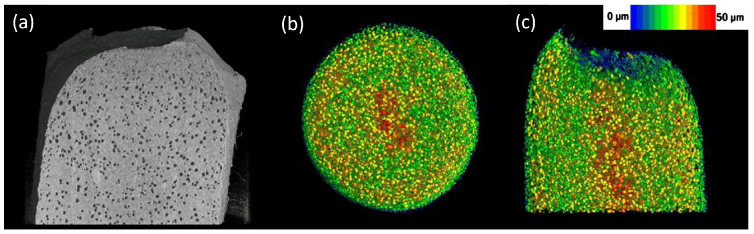
(**a**) Micro tomography 3D image cut through of ethylene vinyl acetate polymer matrix. (**b**) Horizontal and (**c**) vertical cut through of (**a**) where the pore size distribution is rendered within the 3D image. Reused with permission from Ref. [168]. Copyright 2011 Elsevier.

**Figure 3 pharmaceuticals-16-00733-f003:**
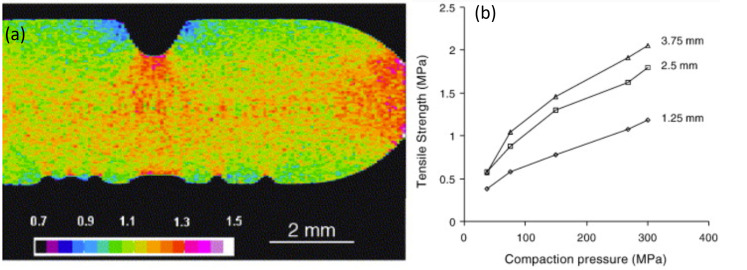
(**a**) Density distribution of microcrystalline cellulose tablet measured by X-ray microtomography. The color scale bar represents the local density expressed in kg/m^3^. Reused with permission from [186]. Copyright 2004 Elsevier. (**b**) Effect of compaction pressure on the tensile strength of compressed aspirin tablets with different thickness. Reused with permission from [51]. Copyright 2009 Elsevier.

**Figure 4 pharmaceuticals-16-00733-f004:**
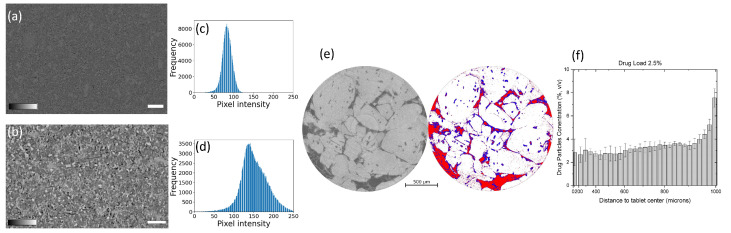
(**a**) X-ray microtomography slice of Janssen Pharmaceutica tablet measured by lab-based X-ray source. The scale bar corresponds to 500 μm. (**b**) Phase contrast X-ray microtomography slice of the same Janssen Pharmaceutica tablet shown in (**a**) measured at Anatomix beamline (Synchrotron SOLEIL) at 40 keV, and the scale bar corresponds to 500 μm. The color scale in (**a**) and (**b**) represents the contrast range from 0 (black) to 255 (white). (**c**,**d**) The histogram representing the grey value distribution of image (**a**,**b**) respectively. (**e**) X-ray microtomography slice of moxidectin tablet measured at TOMCAT beamline (Swiss Light Source) shown in grey scale contrast, and the segmented image representing pharmaceutical ingredients (red-moxidectin, blue-croscarmellose sodium and mixture material, white-functionalized calcium carbonate). (**f**) Representation of volume fraction of moxidectin [red region shown in (**e**) across the radial distance from the center of the tablet. (**e**,**f**) Reused with permission from [188]. Copyright 2020 Elsevier.

**Figure 5 pharmaceuticals-16-00733-f005:**
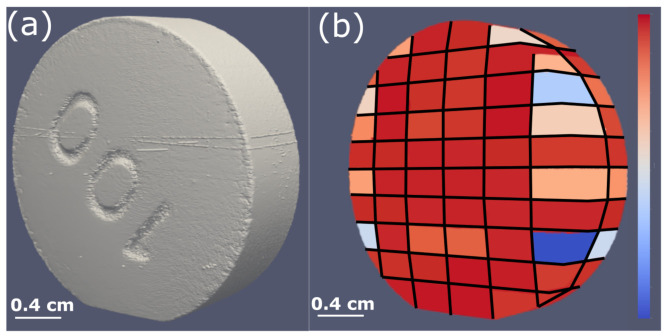
(**a**) Image of tablet volume. (**b**) 3D macropixles of the tablet volume shown in (**a**). The color scale indicates the level of uniformity between two components present in the tablet, red—high uniformity, blue—low uniformity.

**Figure 6 pharmaceuticals-16-00733-f006:**
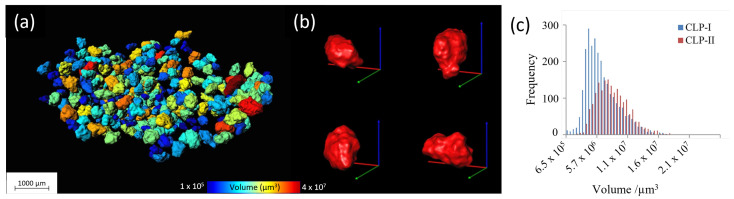
(**a**) Segmented image illustrating the volume distribution of clopidogrel bisulphate particles with the color scale representing the volume of the individual particles in μm^3^. (**b**) Isolated particles of clopidogrel bisulphate extracted from the tomography image shown in (**a**) as red. (**c**) Histogram of the volume distribution of the clopidogrel bisulphate particles existing in two different crystallographic phases, the red and blue legend corresponds to the two different phases synthesised in this work. (**a**–**c**) Reused from [297].

**Figure 7 pharmaceuticals-16-00733-f007:**
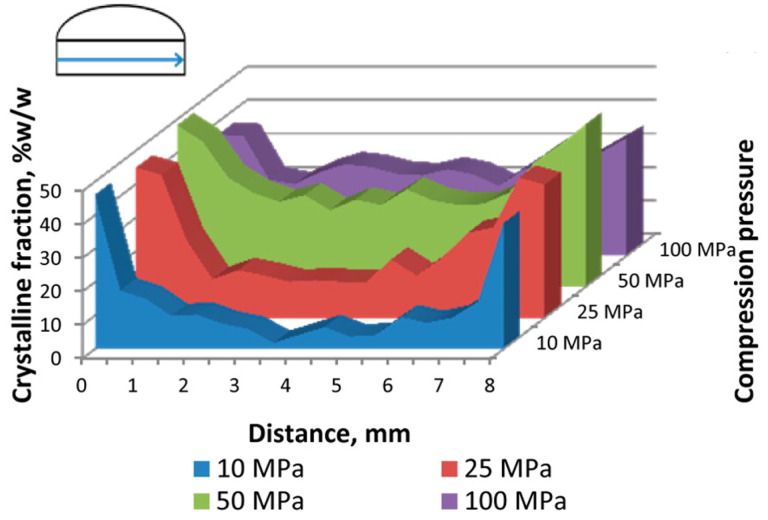
Crystallinity fraction of the amorphous indomethacin tablets measured by 2D X-ray diffraction across the length of the tablet measured at different compressive pressure. The arrow indicates the scanning direction on the tablet. Reused with permission from [306]. Copyright 2015 American Chemical society.

**Figure 8 pharmaceuticals-16-00733-f008:**
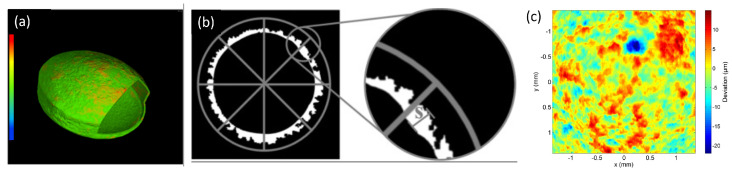
(**a**) Thickness variation of methacrylic acid copolymer LD (L30D-55) and talc coating layer of an asprin tablet analyzed from VGStudioMax (VolumeGraphics GmbH, Heidelberg, Germany), where the colour scale represents the local thickness of the coating layer ranging from 0–150 μm (blue-red). (**a**) Reused with permission from [328]. Copyright 2017 Elsevier. (**b**) Thickness of the coating layer consisting of sodium benzoate, hydroxypropyl methylcellulose and water calculated at certain radial axis marked in the image. Reused with permission from [332]. Copyright 2015 Elsevier. (**c**) Surface roughness profile of tablet coating layer consisting of a mixture of eudragit L30 D-55, triethyl citrate and talc measured by optical coherence tomography. The color scale represents the local thickness variation from the mean thickness of the coating layer. Reused with permission from [247]. Copyright 2017 Elsevier.

**Figure 9 pharmaceuticals-16-00733-f009:**
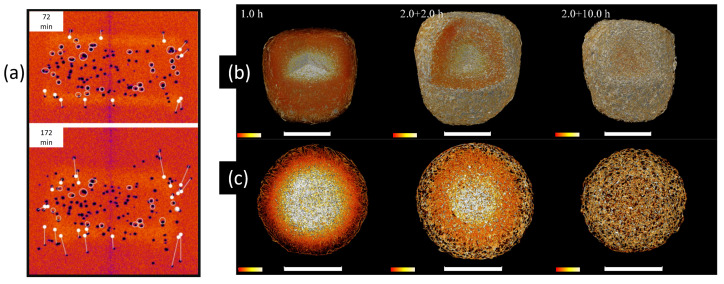
(**a**) Radiography images of the dynamics of a swelling process taken at 72 (**top**) and 172 (**bottom**) min after the start of the dissolution. The tablet consists of microcrystalline cellulose and hydroxypropyl-methyl-cellulose, the dark spots indicates glass microsphere tracers to identify the local swelling movements. The white spots along with the lines indicate the position and the displacement of the tracers from the previous time sequence. Reused with permission from [368]. Copyright 2008 Elsevier. (**b**) 3D reconstructed image measured by phase contrast X-ray microtomography of dried chitosan–λ-carrageenan matrix based tablet during the swelling process. (**c**) The 2D slices of the tablet cross section shown in (**b**) indicating the changes in the tablet matrix during the swelling. The scale bars in (**b**,**c**) correspond to 5000 μm. The polymer network expanded after dissolution of the drug embedded inside the matrix is shown in the last column of (**c**). The white and bright orange region present in the center of the tablet is the drug which was released during the dissolution process. The color scale (ranging from 0 to 255) represents the density variation of the drug, with white color and dark orange color representing high and low density of the drug, respectively. The remaining regions represents the polymer matrix. (**b**,**c**) Reused with permission from [191]. Copyright 2020 Elsevier.

**Figure 10 pharmaceuticals-16-00733-f010:**
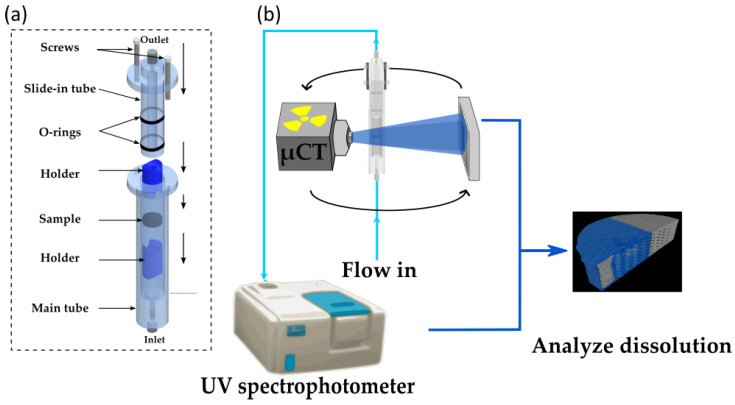
(**a**) 3D model of the flowcell apparatus with the different components. The flow cell is made of polymethyl methacrylate (PMMA) due to its relatively low X-ray attenuation. The dissolution solution is continuously introduced from the bottom inlet tube of the flowcell using a pump. The extracted solution from the outlet can later be used to analyse the dissolved API such as using a UV spectrometer shown in (**b**) to correlate with the microtomography data. (**b**) Schematic representing the experimental setup for 3D characterization of the tablet dissolution using microtomography (EMCT) scanner at UGCT in combination with UV spectrophotometer. The blue color in the reconstructed image of the tablet represents the dissolution solution. (**a**,**b**) is adapted from [187].

**Table 1 pharmaceuticals-16-00733-t001:** Non-limitative list of tablet characterization and the corresponding X-ray tomography methods.

Characterization	Analytical Approaches	Corresponding X-ray Tomography Characterization
Mechanical	Hardness test, acoustic methods, hydrostatic weighing and microscopic examinations, compaction simulator [51,52]	Analyze pore concentration, size, shape, connectivity; measure density distribution
Content identification (assay/uniformity)	High performance liquid chromatography, NIR, IR Raman, vibrational spectroscopy, tablet weighing, hardness test [53,54,55]	Analyze API/structures of interest distribution, density distribution
Intrinsic	Crystallinity-Lab-based diffractometer, small angle X-ray scattering, powder X-ray diffraction [56,57]; particle morphology-SEM	Crystallinity-Diffraction X-ray tomography. Particle morphology-segmentating and analyzing individual particles
Coating layer	Tablet weighing, optical microscopy/SEM	Segment coating layer to analyze local thickness, roughness profile
Dissolution	Disintegration and dissolution testing [58]	Real time disintegration and dissolution imaging

## Data Availability

Not applicable.

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
