# Peer review of "Characterization of Pharmaceutical Tablets by X-ray Tomography"

_pharmaceuticals, 2023, doi:10.3390/ph16050733_

Round 1
Reviewer 1 Report
Characterization of pharmaceutical tablets by X-ray tomography
The paper provides a comprehensive review of all methods that are currently being used and methods under development that may be used in the future to characterize pharmaceutical tablets. The scope of the paper is therefore much broader than only X-ray tomography than the title suggests.
The paper is well-written and easy to read and provide a concise, and as far as I can see, accurate description of all available methods to characterize pharmaceutical tablets. However, since the paper is both broad and concise, there was no room left to discuss the methods in great detail. This means that readers that are interested in a particular method, will have to consult the original papers. The strength of the paper is that it is a catalog of all available methods.
I have a few comments:
Line 286: large coherence lengths (100-200 m). Unfortunately, I do not have access to the original reference, but 100-200 m seems long to me compared to the coherence lengths of about 100 µm I am used to in single crystal diffraction. However, the authors may talk about a different coherence length and they should check that the distance of 100-200 m is indeed correct.
Line 471: taking images at different beam spectra. Here it is not clear to me, what the authors mean. For me a spectrum is what one obtains when one measures e.g. the absorbance as a function of the wavelength. Do they mean that images are being taken at different wavelengths or wavelength ranges? Line 723 suggests that this is the case. This should be clarified in the text. The spectral imaging paragraph is missing citations. It would be good if the authors could give some.
Line 562 tablet characterization. This paragraph is missing citations as well. Since what is written is fairly obvious, it is not essential. Still it would be good if some citations could be given for readers interested in more details.
Line 697-698. Powder and die. I suspect that there is a typo here and that dye is meant.
Line 827: understating the pore network. I assume that understanding is meant.
With the exception of a few potential typo's the manuscript is well written and easy to read.
Author Response
We thank the referee for reviewing the manuscript and providing valuable feedback and suggestions for improving our paper. We have addressed each issue raised by the referees below. The referees’ questions are set in normal font and our response is set in “italics”. Changes made to the manuscript are marked in “blue” for easy identification.
We trust that, with these comments and changes, the manuscript will be found suitable for publication. We thank the referee once again for his/her helpful comments and the journal for giving us the opportunity to further improve our manuscript.
Referee 1:
The paper provides a comprehensive review of all methods that are currently being used and methods under development that may be used in the future to characterize pharmaceutical tablets. The scope of the paper is therefore much broader than only X-ray tomography than the title suggests.
The paper is well-written and easy to read and provide a concise, and as far as I can see, accurate description of all available methods to characterize pharmaceutical tablets. However, since the paper is both broad and concise, there was no room left to discuss the methods in great detail. This means that readers that are interested in a particular method, will have to consult the original papers. The strength of the paper is that it is a catalog of all available methods.
I have a few comments:
- Line 286: large coherence lengths (100-200 m). Unfortunately, I do not have access to the original reference, but 100-200 m seems long to me compared to the coherence lengths of about 100 µm I am used to in single crystal diffraction. However, the authors may talk about a different coherence length and they should check that the distance of 100-200 m is indeed correct.
Response - We thank the reviewer for pointing this out. We wanted to highlight the longitudinal coherence from the insertion device. For example, the energy bandwidth from an undulator can be very narrow, resulting in a larger longitudinal coherence making it possible to place a sample several meters downstream. Since this statement is misleading, we have now re-written this line. (l.272)
“The low divergence enables long propagation distances with both high spatial and high longitudinal (or temporal) coherence from the insertion device to the sample”
- Line 471: taking images at different beam spectra. Here it is not clear to me, what the authors mean. For me a spectrum is what one obtains when one measures e.g. the absorbance as a function of the wavelength. Do they mean that images are being taken at different wavelengths or wavelength ranges? Line 723 suggests that this is the case. This should be clarified in the text. The spectral imaging paragraph is missing citations. It would be good if the authors could give some.
Response – In line 471, we refer the approach carried out on a dual energy micro-CT system. We agree with the reviewer that this statement is misleading when compared with line 723. We have now elaborated this statement and have also added three additional references.(l.447-475)
“To overcome these issues, spectral imaging can be applied by (1) using different source spectra or (2) by using spectral or photon-counting X-ray detectors. In the former, different (yet often overlapping) spectra are used, i.e. different energy ranges as in the case of lab-based dual microtomography setup, such that the ratio in the absorbance signal is different for different chemical components. [159] However, such methods typically have limited efficiency, and require good calibration. The same can also be achieved at the detector level. Spectral imaging/detectors can be divided into multispectral and hyperspectral imaging. In the former, the spectral detectors can measure photons with different energy ranges (or energy bins). They usually have relatively poor energy resolution and suffer from the charge sharing effect, yet promising results have been achieved to identify specific materials. Alternatively, hyperspectral detectors can be used, where the number of energy bins are higher, and can provide higher energy resolution than a multispectral detector. Nevertheless, the energy resolution achievable using a hyperspectral detector is still lower than what is achievable at synchrotrons where the photon energy is tuned by the monochromator, allowing for extremely high spectral resolutions (down to eV level at hard X-ray range). Implementing a hyperspectral detector system can have numerous challenges, and typically require large upgrades at the detector, such as with the electronics and data acquisition software.”
- Line 562 tablet characterization. This paragraph is missing citations as well. Since what is written is fairly obvious, it is not essential. Still it would be good if some citations could be given for readers interested in more details.
Response - We have now added a few citations in the revised version.
- Line 697-698. Powder and die. I suspect that there is a typo here and that dye is meant.
Response - It is indeed confusing, but “die” is the correct word for a die and punch method for tableting. Instead of using simply “die”, now we refer them as “die and punch”.
- Line 827: understating the pore network. I assume that understanding is meant.
Response - We apologies for this error. It is now corrected.

Reviewer 2 Report
The review has a relatively comprehensive theme that focuses on the characterization of solid formulations through X-ray computed tomography, and introduces the development and application of this technology. It is relevant, with appropriate references cited. It is recommended that the opening section of the Introduction (1-71) be streamlined to quickly get to the main topic.
Author Response
We thank the referee for reviewing the manuscript and providing valuable feedback and suggestions for improving our paper. We trust that after this correction, the manuscript will be suitable for publication.
Response - We agree with the reviewer and have removed the unnecessary texts in lines 1-71, and paragraphs are highlighted in blue for easy identification.
Reviewer 3 Report
The MS entitled "Characterization of pharmaceutical tablets by X-ray tomography" authored by Jaianth Vijayakumar, Niloofar Moazami Goudarzi, Guy Eeckhaut, Koen Schrijnemakers, Veerle Cnudde, Matthieu N Boone is well written and organized, The manuscript could be accepted for publication after minor revision
1. Addition of one or two tables will enhance the readership
2. In-depth discussion and a concluding statement should be included in each subheadings
3. Minor english correction is required (spell check & typo errors)
Author Response
We thank the referee for reviewing the manuscript and providing valuable feedback and suggestions for improving our paper. We have addressed each issue raised by the referees below. The referees’ questions are set in normal font and our response is set in “italics”. Changes made to the manuscript are marked in “blue” for easy identification.
We trust that, with these changes, the manuscript will be found suitable for publication. We thank the referee once again for his/her helpful comments and the journal for giving us the opportunity to further improve our manuscript.
The MS entitled "Characterization of pharmaceutical tablets by X-ray tomography" authored by Jaianth Vijayakumar, Niloofar Moazami Goudarzi, Guy Eeckhaut, Koen Schrijnemakers, Veerle Cnudde, Matthieu N Boone is well written and organized, The manuscript could be accepted for publication after minor revision
- Addition of one or two tables will enhance the readership
Response – We agree with the reviewer, and we have added a table summarizing the different X-ray modalities, resolution and the measurement time. (p.13)
- In-depth discussion and a concluding statement should be included in each subheadings
Response – We have now added a short discussion and a concluding statement in each subheadings where it is missing and highlighted the places where it already exist (all now highlighted in blue). We also took the opportunity to include an extra image supporting our discussion. (p.19)

Reviewer 4 Report
This is a good review on visualization of pharmceutical tablets by X-ray CT, which will be worth publishing in this journal. I would suggest one minor comment for revion. It would be good to add list of table to compare typical X-ray CT methods reported herein in terms of time resolution and spatial resolution.
Author Response
We thank the referee for reviewing the manuscript and providing valuable feedback and suggestions for improving our paper. We trust that with this correction, the manuscript will be suitable for publication.
Response – We agree with the reviewer and have added this table in the revised version of the manuscript. (p.13)